# Composition Effect on the Formation of Oxide Phases by Thermal Decomposition of CuNiM(III) Layered Double Hydroxides with M(III) = Al, Fe

**DOI:** 10.3390/ma17010083

**Published:** 2023-12-23

**Authors:** Iqra Zubair Awan, Phuoc Hoang Ho, Giada Beltrami, Bernard Fraisse, Thomas Cacciaguerra, Pierrick Gaudin, Nathalie Tanchoux, Stefania Albonetti, Annalisa Martucci, Fabrizio Cavani, Francesco Di Renzo, Didier Tichit

**Affiliations:** 1ICGM, University Montpellier, CNRS, ENSCM, 1919 Route de Mende, 34090 Montpellier, France; iqrazubair@gmail.com (I.Z.A.); phuoc@chalmers.se (P.H.H.); bernard.fraisse@umontpellier.fr (B.F.); thomas.cacciaguerra@enscm.fr (T.C.); pierrick.gaudin@umontpellier.fr (P.G.); nathalie.tanchoux@enscm.fr (N.T.); francesco.di-renzo@umontpellier.fr (F.D.R.); 2Department of Industrial Chemistry “Toso Montanari”, Alma Mater Studiorum-Università di Bologna, Viale Risorgimento 4, 40136 Bologna, Italy; stefania.albonetti@unibo.it (S.A.); fabrizio.cavani@unibo.it (F.C.); 3Department Chemistry, Lahore Garrison University, Lahore 54000, Pakistan; 4Chemical Engineering, Competence Centre for Catalysis, Chalmers University of Technology, SE-412 96 Gothenburg, Sweden; 5Department Physics and Earth Sciences, University of Ferrara, Via Saragat 1, 44100 Ferrara, Italy; bltgdi@unife.it (G.B.); mrs@unife.it (A.M.); 6Center for Chemical Catalysis—C3, Alma Mater Studiorum-Università di Bologna, Viale Risorgimento 4, 40136 Bologna, Italy

**Keywords:** layered double hydroxides, thermogravimetry, in situ X-ray diffraction, tenorite, bunsenite, spinel aluminate, spinel ferrite

## Abstract

The thermal decomposition processes of coprecipitated Cu-Ni-Al and Cu-Ni-Fe hydroxides and the formation of the mixed oxide phases were followed by thermogravimetry and derivative thermogravimetry analysis (TG – DTG) and in situ X-ray diffraction (XRD) in a temperature range from 25 to 800 °C. The as-prepared samples exhibited layered double hydroxide (LDH) with a rhombohedral structure for the Ni-richer Al- and Fe-bearing LDHs and a monoclinic structure for the CuAl LDH. Direct precipitation of CuO was also observed for the Cu-richest Fe-bearing samples. After the collapse of the LDHs, dehydration, dehydroxylation, and decarbonation occurred with an overlapping of these events to an extent, depending on the structure and composition, being more pronounced for the Fe-bearing rhombohedral LDHs and the monoclinic LDH. The Fe-bearing amorphous phases showed higher reactivity than the Al-bearing ones toward the crystallization of the mixed oxide phases. This reactivity was improved as the amount of embedded divalent cations increased. Moreover, the influence of copper was effective at a lower content than that of nickel.

## 1. Introduction

Layered double hydroxides (LDHs) and the derived mixed oxides obtained by thermal decomposition attract tremendous interest as adsorbents, catalysts, drug carriers, additives of polymers, and the material of electrodes [1,2,3,4,5,6,7,8]. This is mainly due to the diversity of compositions, which offers a unique opportunity to tune the nature, strength and distribution of the active sites. Moreover, the nature of the mixed oxides obtained from each type of LDH precursor tightly depends on the calcination temperature and can be thus finely tailored.

The general chemical formula of LDHs can be written as [M(II)_1−x_M(III)_x_(OH)_2_][A^n−^_x/n_]·mH_2_O, where M(III) cations are typically Al, Cr, Fe, Ga, and M(II) cations, typically Mg, Zn, Ni, Co, and Cu, that occupy the centres of M(OH)_6_ octahedral units sharing edges to form positively charged two-dimensional brucite-type sheets. A^n−^ is an exchangeable charge-compensation anion; x generally between 0.20 and 0.40 represents the M(III) molar fraction referred to metals. Stacking of the brucite-type layers can be accomplished in two ways, leading to two polytypes with a 3R rhombohedral symmetry or a 2H hexagonal cell.

The number of different octahedrally coordinated cations that can be introduced in the brucite-like layers is theoretically unlimited as long as the ionic radius is in the range of 0.65–0.80 Å and 0.62–0.69 Å for the trivalent and divalent cation, respectively, although the most common trivalent Al^3+^ is significantly smaller (0.50 Å). However, cations with larger ionic radii and/or higher coordination, such as lanthanides or noble metal cations as well as tetravalent cations (e. g. Zr^4+^, Sn^4+^, and Ti^4+^), have been introduced in M^2+^M^n+^-LDHs or in multicationic LDHs containing a large number of divalent and/or trivalent cations [9,10,11,12,13,14,15,16,17,18,19].

Focusing on the catalytic applications, they were for a long time mainly developed with LDHs, combining in the layers a couple of divalent and trivalent cations able to provide a highly efficient acid–base and supported metal catalysts for condensation, hydrogenation, and oxidation reactions, as already extensively reviewed [20,21,22,23,24,25,26]. Nowadays, a huge amount of acido-basic and multifunctional catalysts achieving complete sequences of a multistep, one-pot cascade or tandem reactions are currently developed using multicationic LDH precursors [20,27,28,29,30,31,32]. Indeed, a majority of LDH-based catalysts, used in one-pot bifunctional reactions for C-C and C=C bond formation, biomass conversion, decomposition of volatile organic compounds (VOCs), DeNOx and DeSOx processes, or photocatalysis, are ternary or quaternary LDHs [20,24,27]. CuZnAl-LDHs are among the first multicationic LDHs extensively studied as precursors of catalysts for the synthesis of methanol from syngas, the water–gas shift reaction, the steam reforming of methanol and ethanol, and the partial oxidation of methanol for hydrogen production [33,34,35,36,37,38]. Lastly, multicationic LDHs were particularly involved in the transformation of biomass-derived model molecules and the lignocellulosic feedstock biorefinery [20,27].

It is noteworthy that several recent works using mixed oxide catalysts obtained from ternary or quaternary LDH precursors were based on the combination of Ni, Cu, Fe, and/or Al cations, with sometimes Mg. Several examples deserve much attention. NiMgFeAl-LDHs with different Ni/Fe molar ratios have been used as precursors of Ni-Fe alloys supported on MgAl_2_O_4_ obtained by calcination at 500 °C of the LDH followed by reduction at 700 °C of the mixed oxides to achieve direct ethanol synthesis (DES) from syngas. Such alloys give rise to a synergetic effect between Fe active for both CO dissociative adsorption and carbon-chain growth, and Ni promoting CO insertion [39]. CuNiMgAl-LDHs with different Cu/Ni molar ratios directly reduced between 400 and 750 °C have been used to prepare Cu-Ni alloys supported nanocatalysts for the selective hydrogenation of furfural into tetrahydrofurfuryl alcohol and furfuryl alcohol [40]. CuNiFe-LDHs with different Cu/Ni molar ratios calcined in the temperature range from 300 to 500 °C have been used as precursors of catalysts for the oxidation of sulfur-containing VOC, particularly CH_3_SH to SO_2_ [41]. Ni and Fe improve the ability of electron transfer and promote the adsorption of CH_3_SH while Cu brings oxygen surface species. NiMgFeAl-LDH precursors with a fixed Ni/Fe ratio and variable Mg/Al ratios reduced at 600 °C were investigated as catalysts for CO_2_ methanation [42]. The activity and CH_4_ selectivity result from an intermediate density of metallic and basic sites with a synergistic effect.

Our group has recently used mixed oxide catalysts derived from CuNiAl-LDH and CuNiFe-LDH precursors with different Cu/Ni ratios calcined at 600 °C for the conversion of a lignin model molecule, 4-benzyloxy-3-methoxybenzaldehyde (BMBA), by catalytic hydrogen transfer from methanol and ethanol at a temperature and pressure similar to the ones of organosolv pulping processes [43,44]. Several types of reactions, i.e., Merwein–Pondorf–Verley (MPV) hydrogenation, hydrogenolysis, acetalisation and condensation were observed with selectivities depending on the cationic composition of the catalysts. Copper provided more active sites and a synergistic effect with Ni improving its reducibility and then hydrogenation activity. Moreover, we established a strong influence on the nature of the trivalent cation, which can be related to structural features such as the mean average size and amounts of CuO and NiO phases.

All the previous works emphasize the versatile character of the Cu, Ni, Fe, and/or Al-bearing mixed oxide catalysts obtained from ternary or quaternary LDH precursors able to achieve various reactions. It is noteworthy that, for binary LDHs, extensive investigations of the decomposition processes and the nature and content of the crystallographic phases upon thermal activation have been achieved. They reveal particular behaviours of LDH such as the “memory effect”, the topotactic decomposition into poorly crystallized mixed oxides, and the reconstruction ability of the MgAl mixed oxides into a layered structure [45,46,47,48,49].

In spite of the dramatic interest in the ternary CuNiAl- and CuNiFe-LDHs, a comprehensive study of the decomposition processes and the nature and ratios of the mixed oxide phases obtained in a large range of temperatures has not yet been performed. A better understanding of the topotactic decomposition and the crystallization processes of the CuNiAl- and CuNiFe-bearing mixed oxides, the formation thresholds, and the kinetics for the growth of the different crystalline phases will provide relevant insights for further applications of these materials. To get deep knowledge of these features, in this work, TG–DTG and in situ XRD analyses were performed, with CuNiAl- and CuNiFe-bearing materials obtained by coprecipitation in alkaline media with different Cu/Ni molar ratios and thermally treated in a temperature range from 25 to 800 °C.

## 2. Materials and Methods

Nickel (II) nitrate hexahydrate ≥ 98.5% (Ni(NO_3_)_2_.6H_2_O), copper (II) nitrate trihydrate ≥ 99% (Cu(NO_3_)_2._3H_2_O), aluminium nitrate nonahydrate ≥ 98% (Al(NO_3_)_3_.9H_2_O), iron (III) nitrate nonahydrate ≥ 99.95% (Fe(NO_3_)_3_.9H_2_O), and sodium hydroxide ≥ 97% (NaOH) from Sigma Aldrich (Burlington, Ma, USA) and sodium carbonate ≥ 99% (Na_2_CO_3_) from VWR (Radnor, PA, USA) was purchased and used as received without any further purification. Deionized water was used throughout the synthesis experiments.

Five samples with a cation ratio x = ΣM(II)/M(III) = 3 and divalent cation ratios such as y = Cu/(Cu + Ni) = 0, 0.1, 0.5, 0.9, 1, were prepared for each of the trivalent cations (Al or Fe). For each synthesis, 100 mL of a solution 0.38 M in divalent cations and 0.125 M in trivalent cation was prepared with the appropriate amounts of the precursor nitrate salts and added dropwise to 100 mL of a Na_2_CO_3_ solution (0.5 M) at the rate of 0.05 mL/s with constant stirring at 1000 rpm. All syntheses were carried out at 30 °C with a controlled addition of alkaline solution NaOH (2 M) by pH-STAT Metrohm 877 Titrino, maintaining the pH at 10 ± 1. The resulting slurry was aged for 15 h at 80 °C under constant stirring at 1000 rpm, centrifuged at 5000 rpm for 10 min, and washed three times with deionized water to remove the excess alkali. The samples were dried under a vacuum for 3 h and further kept in an oven at 80 °C for 12 h. Samples are named by the percent atomic fraction of cations, for instance, Cu38Ni37Fe25.

X-ray patterns were recorded with Cu Kα radiation on a PANalytical Empyrean diffractometer with a chamber Anton-Paar HTK16 (Malvern Panalytical Ltd., Malvern, United Kingdom) at a heating rate of 5 °C/min from 25 to 800 °C with acquisition every 100 °C. The powder samples were deposited on PtRh supports. Thermogravimetric analysis was carried out with a Perkin Elmer STA 6000/8000 apparatus (Wellesley, MA, USA). Approximately 25 mg of catalyst were heated in an airflow of 60 mL/min to 800 °C with a linear heating rate of 5 °C/min. The cation ratio of the prepared samples was measured by energy dispersive X-ray analysis (EDX) with a QUANTA 200F instrument with detector Oxford Instruments X-Max N SDD (Abingdon-on-Thames, United Kingdom). The patterns of the detected phases are Ni_6_Al_2_(OH)_16_CO_3_·4H_2_O (JCPDS 15-0087), NiAl_2_O_4_ (JCPDS 10-0339), Cu_6_Al_2_(OH)_16_CO_3_·4H_2_O (JCPDS 37-0630), CuO tenorite (JCPDS 41-0254), and CuAl_2_O_4_ (JCPDS 01-1153), albeit the diffraction peaks were often shifted by solid solution and thermal expansion effects.

## 3. Results

The structural evolution of the two (Cu,Ni)Al and (Cu,Ni)Fe series of materials, with a cation ratio x = ΣM(II)/M(III) = 3 and divalent cation ratios y = Cu/(Cu + Ni) = 0, 0.1, 0.5, 0.9, and 1, were analyzed in the following by TG–DTG and in situ XRD analysis.

### 3.1. (Cu,Ni)Al Series

The XRD patterns of the as-synthesized samples of the (Cu,Ni)Al series are reported in Figure 1. Those of the samples with 0 ≤ y < 0.9 are characteristic of trigonal LDH structure with 00l reflections in the 2θ range below 25° and 0kl reflections above 30°. The patterns of the samples with 0.9 ≤ y ≤ 1 (Cu68Ni07Al25, Cu75Al25) instead of a trigonal system are fitted in a monoclinic system due to Jahn–Teller distortion of the CuO_6_ octahedra, as previously pointed out by Yamaoka [50]. In addition, a slight amount of tenorite is present for y = 0.9. The basal spacings of the LDH phases are reported in Table 1. They depend on the size of the intercalated anion, the amount of water molecules, and the charge density of the layers, which determine the electrostatic interaction with the anions. The basal spacing varies between 7.61 to 7.52 Å and is in agreement with the intercalation of CO_3_^2−^ anions. It must be pointed out that, in both series, Al- and Fe-bearing LDHs, CO_3_^2−^ is the only compensating anion, as coprecipitation has been performed in the presence of Na_2_CO_3_. The slight decrease of the interlayer space when y increases agrees with the decrease in the amount of hydration water in the LDH, as revealed by TG analysis (see Table 2).

The TG-DTG profiles of the LDH structures, reported in Figure 2, exhibit different shapes. The samples with y ≤ 0.5 show three mass losses corresponding to endothermic effects (Table 2). This is in close agreement with the decomposition process largely described for LDH materials [51,52,53]. The first mass loss is generally assigned to the release of weakly bonded water molecules from the surface and the interlayer, with a component of partial dehydroxylation of the layers. The second mass loss is due to the elimination of thermally decomposed anions, such as carbonates, and water molecules resulting from the condensation of the hydroxyls. The third mass loss corresponds to the decomposition of residual carbonates.

Considering then the LDHs with y > 0.5, the TG profiles of Cu68Ni07Al25 and Cu75Al25 exhibit four different mass losses. The behaviour of these copper-richer samples is in agreement with the complex decomposition process generally reported for CO_3_-CuAl-LDH, with the two first steps assigned to dehydration and dehydroxylation, followed in the higher temperature range by several steps attributed to the decomposition of intermediately formed carbonate species. Hence, decomposition processes with two to five mass loss steps have been previously reported for these CO_3_-CuAl-LDHs [54,55,56,57,58].

The first decomposition step, assigned to dehydration, is shifted toward lower temperature as the copper content increases, with the end of the mass loss moving from 235 to 167 °C (Table 2). It must be noted that for the Ni-richer samples, in spite of a continuous mass loss in the first decomposition step, the corresponding DTG peak (DTG_1_ in Table 2) is asymmetric, with several maxima toward lower temperatures. These features account for the removal of weakly bonded water molecules adsorbed on the surface and of interlayer water molecules, hydrogen-bonded to carbonate and to hydroxyl groups.

The second DTG peak (DTG_2_) is also moving toward lower temperatures as the copper content increases, showing easier dehydroxylation, although it becomes a more complex feature being concomitant with partial dehydration and decarbonation, as will be further detailed. However, this behaviour is consistent with the easier dehydroxylation of Cu(OH)_2_ than Ni(OH)_2_ into the respective oxides, occurring at ~150 and ~300 °C, respectively [59,60].

The nature and formation thresholds of the crystallographic phases during the decomposition processes are obtained from in situ XRD. The patterns, recorded every 100 °C during continuous heating from 25 to 800 °C, are reported in Appendix A.

The characteristic pattern of the LDH phase is observed between 25 and 200 °C in all samples, with a shift toward higher 2θ values of the 00l reflections (Table 1) and a concurrent peak broadening. Therefore, dehydration occurring in this temperature range gives rise to a shrinkage of the interlayer space and a decrease in crystallinity, showing that the structure becomes partially disordered [51].

The interlayer space for the two Ni richer samples (Ni75Al25 and Cu07Ni68Al25) (Table 1) decreases very slightly (0.22–0.24 Å) between 25 and 100 °C, where it reaches 7.39 Å, showing that water molecules are removed but carbonates remain in the same position and symmetry [61]. The interlayer space decreases much more above 100 °C to reach ca. 6.55 Å at 200 °C. The same large decrease of the interlayer space to reach 6.52–6.62 Å is observed since 100 °C for copper-richer samples with 0.5 ≤ y ≤ 0.9. Therefore, the major structural change is shifted from 200 to 100 °C and gives a larger collapse of the interlayer space (from ca. 1.0 to 1.9 Å) as the copper content increases, in agreement with the shift to a lower temperature of the DTG_1_ peak. The shrinkage of the interlayer space reaches 1.9 Å at 200 °C for Cu68Ni07Al25 and Cu75Al25. It is noteworthy that Cu75Al25 has a peculiar behaviour with the simultaneous presence at 100 °C of the two phases, i.e., the simply dehydrated one giving an interlayer space of 7.49 Å and the one with an interlayer space of 6.31 Å with shrinkage of ca. 1.2 Å (Figure 3).

The nature of the observed phase with large shrinkage has given rise to controversies. Bellotto et al. have simulated the XRD pattern of a MgAl-LDH (Mg/Al = 2) dehydrated at 200 °C, showing a collapse of the interlayer space from 7.59 to 6.57 Å and about 10% of the Al becoming tetrahedrally coordinated. Their model fits with a structure where some Al atoms diffuse from the brucite-like layers toward the interlayer domain [45]. These tetrahedrally coordinated aluminium atoms are surrounded by three oxygens of the layer and one apical oxygen in the interlayer. A vacant site is created in the brucite-like layer. Kanezaki, on the other hand, has described the formation of a thermally metastable phase between 180 and 380 °C in CO_3_-MgAl-LDH with an atomic ratio of Mg/Al = 3 [62]. It is indexed as a hexagonal phase with an interlayer space of 1.8 Å and exhibits low layer stacking, with only two 00l reflections. It is assumed that, during the formation of this phase upon thermal treatment, hydroxyl anions are produced by the interaction of carbonates with interlayer water molecules. The replacement of the compensating carbonates by the hydroxyls accounts for the decrease of the interlayer space from ca. 7.8 Å to 6.6 Å and the poor stability of the phase. However, Rives pointed out that an interlayer space of 6.6 Å does not match with that to be expected for the intercalation of hydroxyls with an ionic radius of 1.19–1.23 Å. The resulting interlayer space indeed would be 7.18–7.26 Å, larger than that found and not significantly different than with carbonate [63]. Rives’s hypothesis is that, upon calcination above 250 °C, the decrease of the interlayer space and the simultaneous removal of carbonate account for the conversion of some hydroxyls of the layers into oxide ions by interaction with CO_3_^2−^. Staminirova et al., in the case of MgAl-LDH heated at 160–180 °C, found a decrease from 7.84 to 6.60 Å of the interlayer space, with preservation of the octahedral environment of Al and Mg, partial dehydroxylation and bidentate bonding of the CO_3_^2−^, with change of symmetry from D_3h_ to C_2v_ [61]. Based on these results, they suggested that the obtained phase results from the bigrafting of carbonate to the layers. The interlayer space of 6.60 Å, too small to accommodate the carbonate anions in a flat position, shows that they are incorporated through two of their oxygens to the layers, with the third oxygen remaining in the interlayer. In this structure, the CO_3_^2−^ environment is similar to that in the natural mineral dawsonite, NaAl(CO_3_)(OH)_2_. This model has been disputed by Vaysse et al. because the distance between two first neighbour oxygen atoms in the layers of about 3 Å is not consistent with the distance between the oxygens in the CO_2_ species in the range from 2.08 to 2.30 Å [64]. Alternatively, Vaysse et al. suggest a monografting of the carbonate anions through the replacement of one hydroxyl of the layer and bonding of one oxygen of the carbonate to the accessible cation. This leads to NiFe- and NiCo-LDH interlayer spaces of 6.5–6.7 Å, which are very close to the experimental ones.

In the LDHs of this study, there is a clear influence of both the copper content (y) and the nature of the trivalent cation on the interlayer space and the formation threshold of the two stages appearing between 25 and 200 °C. The phase present at a lower temperature with low shrinkage of the interlayer space (0.22–0.24 Å) corresponds only to dehydration, without changes in the carbonate symmetry. The phase present at higher temperatures, with a larger decrease of the interlayer space (0.9–1.1 Å), accounts for a more significant structural transformation. It must be pointed out that it appears at a lower temperature with a larger decrease of the interlayer space in the copper-richest samples. We have previously shown that the later ones are more easily dehydroxylated than the nickel-richer samples. We suggest that the mechanism described by Vaysse et al., with partial dehydroxylation and monografting of the carbonates, likely takes place in the Ni-richest samples (Ni75Al25 and Cu07Ni68Al25) with moderate shrinkage of ca. 1.1 Å and no decarbonation below 200 °C. Accordingly, this mechanism has been also observed in NiCo-, NiFe and NiAl-LDHs [64]. The mechanism described by Bellotto et al. [45] more likely accounts for the behaviour of the Cu-richest samples, where, due to the Jahn–Teller effect, the rearrangement of the brucite-like layers with the migration of aluminium to the interlayer can more easily occur. Easier dehydroxylation is another aspect of the same phenomenon. This mechanism can also account for the larger shrinkage of the interlayer space, reaching ca. 1.9 Å.

The mass fractions of the oxide phases formed between 300 and 800 °C are reported in Appendix A, together with their Scherrer crystallite size. In order to highlight the main steps of evolution, the representative patterns recorded at 400, 600, and 800 °C, i.e., respectively after complete decomposition of the LDH, at an intermediate temperature and after complete recrystallization, are brought together in Figure 4. The samples contain crystalline NiO, CuO, and spinel phases, whose ratios depend on composition and temperature (Appendix A).

Ni75Al25 features the (111), (200), and (220) reflections of NiO bunsenite at, respectively, 37.3, 43.3, and 62.7° 2θ, throughout the temperature range from 300 to 800 °C (Figure 4A). The position of the reflections slightly shifts at a lower 2θ angle with the thermal expansion of the sample at a rising temperature. It is noteworthy that bunsenite appears at 300 °C, a temperature almost corresponding to the endothermic peak assigned to dehydroxylation and decarbonation (DTG_2_) at 321 °C (Table 2), although TG–DTG and XRD experiments are performed in different conditions. A different relevant reflection at 35.4° 2θ appears at 300 °C and retains a similar intensity at a higher temperature. Such a reflection, present with lower intensity also in Cu07Ni68Al25 and Cu38Ni37Al25, has already been observed in the calcination of NiAl LDH and attributed to the (311) peak of a not better-defined spinel phase [65]. However, this attribution seems questionable, as the reflection would correspond to a spinel with a = 8.40 Å, too large for any aluminate spinel [66], and the peak more likely pertains to a defective alumina.

A quantitative distribution of Ni^2+^ and Al^3+^ cations in the different phases of Ni75Al25 is difficult to ascertain. XRD analysis indicates that the mass fraction of bunsenite, already 40% at 300 °C, rises after 500 °C and reaches 57% at 800 °C (Appendix A).

The Scherrer evaluation of crystallite size can be heavily affected by the presence of crystal defects and strain, a common occurrence in nanocrystals formed at a relatively low temperature. However, the evolution of the Scherrer crystallite size of NiO, from nearly 2.5 to 4 nm between 500 and 800 °C (Appendix A), corresponds to the expected effect of temperature on crystal growth and, possibly, better ordering.

The Cu38Ni37Al25 sample, even with content of copper and nickel (Figure 4B), presents at 400 °C a very limited amount of crystallized NiO bunsenite and CuO tenorite (9 and 4% mass, respectively, see Appendix A), showing a much more slugger crystallization than Ni75Al25. Above 400 °C, the amount of crystalline phases increases, reaching 39% NiO and 36% CuO at 800 °C. In the meantime, between 500 and 800 °C, the Scherrer crystallite size increases, from 5 to 11 nm for NiO and from 10 to 25 nm for CuO. No evident aluminium-bearing phase is detected.

Also, Cu75Al25 presents a small amount of crystallized CuO tenorite at 400 °C (Figure 4C). Above 500 °C, the amount of tenorite and its crystallite size increase (Appendix A). At 700 °C, a spinel phase appears, and, at 800 °C, the sample is completely crystallized, with 62 and 37 mass % of, respectively, tenorite and spinel, which is in good agreement with the chemical composition. The spinel phase presents a cell size of a = 8.072 Å, to be compared with a literature value of 8.086 Å for CuAl_2_O_4_ [66].

### 3.2. (Cu,Ni)Fe Series

The XRD patterns of the as-synthesized (Cu,Ni)Fe samples are reported in Figure 5. For the Ni-rich samples Ni75Fe25 and Cu07Ni68Fe25, rhombohedral LDH is the only phase present. In Cu38Ni37Fe25, the LDH pattern is accompanied by two CuO tenorite reflections at 35.5 and 39° 2θ. At a higher copper content, the intensity of the CuO reflections increases. Only very weak 003 and 006 LDH reflections are observed for Cu68Ni07Fe25, and no LDH is visible for Cu75Fe25, which presents only broad tenorite reflections. It can be recalled that the difficult formation of a CuFe-LDH phase has been often reported [67,68]. It has been explained by the different pH of precipitation of hydroxylated Cu^2+^ and Fe^3+^ precursors, hindering the coprecipitation of a common Cu-Fe hydroxide phase [44].

It can be observed that the interlayer space of the LDH phase is larger in Fe- than Al-bearing samples of the same composition due to the lower charge density of Fe^3+^ than Al^3+^, which decreases the electrostatic interaction with the carbonate anions (Table 1). The cell parameter of the LDH structures, calculated from the position of the 110 reflections at 60–61° 2θ, shows lower values in the aluminium- than iron-bearing samples, which is in agreement with the lower ionic radius of Al^3+^ (0.50 Å) than Fe^3+^ (0.64 Å).

The TG profiles of the LDH samples with 0 ≤ y ≤ 0.5 show main mass losses in the temperature ranges of 25–200 and 200–350 °C (Figure 6) (Table 2). As in the case of the corresponding (Cu,Ni)Al samples, these phenomena correspond, respectively, to dehydration and dehydroxylation–decarbonation.

The TG-DTG profiles of Cu75Fe25, exhibiting just a tenorite crystal phase, have obviously a different general shape than the LDH samples, and show three shallow mass losses with DTG peaks at 77, 190, and 427 °C and a total mass loss of 12%. These peaks can account for the presence of iron and/or copper hydroxide and/or carbonate phases. It has been shown that Fe(OH)_3_ dehydration occurs at ~260 °C [69] and decompositions of Cu_2_CO_3_(OH)_2_ malachite at 230–310 °C and FeCO_3_ siderite at ~450 °C [55,70]. For the decomposition of goethite, FeOOH, the mass loss from 25 to 172 °C was assigned to the release of physically adsorbed water molecules, and the one between 172 and 310 °C to dehydroxylation resulting in the transformation of goethite to hematite [71]. The weak mass loss (1.3%) occurring between 310 and 1000 °C in Cu75Fe25 is due to the decomposition of remnant structural hydroxyl and partly to nonstoichiometric hydroxyl units. The TG-DTG profile of Cu68Ni07Fe25 is similar to that of Cu75Fe25, with the addition of a 4% mass loss peaking at 143 °C, corresponding to the dehydration of a low amount of the LDH phase.

In correspondence with the dehydration peak, a different evolution of the interlayer space is observed for the Fe- and Al-bearing LDH series (Table 1). For the two Ni-richest samples, the shrinkage of the interlayer space upon heating at 100 °C is about 0.2 Å in the (Cu,Ni)Al series and 0.6–1 Å in the (Cu,Ni)Fe series, confirming the easier dehydration of LDH containing a larger trivalent cation in the layers.

The XRD patterns of the formation of crystalline oxide phases, recorded every 100 °C during continuous heating up to 800 °C, are reported in Appendix A. The mass fraction and the Scherrer crystallite size of the crystal phases are reported in Appendix A. As previously done for the (Cu,Ni)Al series, the representative patterns at 400, 600, and 800 °C are brought together in Figure 7.

The XRD pattern of Ni75Fe25 (Figure 7A) presents only the (111), (200), and (220) reflections of NiO bunsenite at, respectively, 36.9, 43.1, and 62.5° 2θ until 600 °C. The formation of bunsenite at 300 °C is consistent with the temperature of the endothermic peak assigned to dehydroxylation and decarbonation at 294 °C (DTG_2_ in Table 2). The bunsenite reflections are quite broad and correspond to the Scherrer crystallite size of 2.4 nm at 400 °C (Appendix A). With the increase in temperature, the NiO peaks become narrower, and the crystallite size increases up to 7.4 nm at 800 °C. In the meantime, the (220), (311), and (511) reflections of a spinel phase appear at 700 °C. The spinel phase, with a cell size of a = 8.325 Å, comparable to the literature value of 8.339 Å for NiFe_2_O_4_ trevorite [66], reaches 12 mass % at 800 °C, whereas NiO decreases from 65 to 45 mass %. At any temperature, crystalline phases account for no more than two-thirds of the mass of the sample, indicating a high stability of the amorphous material formed by LDH decomposition.

At higher copper content, both NiO bunsenite and CuO tenorite are present. In Cu38Ni37Fe25, their sum represents nearly one-third of the mass of the sample at 300 °C but steadily increases with the rise of temperature (Appendix A). At 600 °C, a spinel phase appears, with the main (311) reflection superposed to the (002) reflection of tenorite at 35.5° 2θ. At 800 °C, the sample is completely crystallized, with spinel representing 37 mass %.

The Cu75Fe25 sample presents broad (002) and (111) reflections of tenorite since 300 °C (Figure 7C). Tenorite represents about 50 mass % of the sample until 500 °C. At a higher temperature, the amount of tenorite barely increases, but a spinel phase appears at 600 °C and represents as far as 25 mass % of the sample at 800 °C. The cell size of the spinel phase is 8.40 Å, higher than the literature value of 8.369 Å for cuprospinel CuFe_2_O_4_ and likely corresponding to a nonstoichiometric iron-rich copper ferrite [72].

## 4. Discussion

### 4.1. Composition Effects in the Decomposition of LDHs

The observations reported in the results section provide several hints on the ways in which the changes of composition influence the stability of LDHs and the properties of the oxides issued from their decomposition. The effect of the cationic nature and content on the decomposition of LDHs can be followed by comparing the DTG curves of samples with different Cu/Ni ratios and different trivalent cations, Al or Fe. Figure 8 reports the DTG curves of Fe- or Al-bearing samples with Ni as only divalent cation (Figure 8A) or even contents of Cu and Ni (Figure 8B).

The temperature of the maxima of DTG peaks is reported in Table 2. The DTG_1_, mainly attributed to the release of interlayer water molecules, presents a main peak at 170–180 °C for y = 0 and either Al or Fe trivalent cation (Figure 8A). The temperature of the peak decreases with the increase of y and reaches 153 °C for Cu38Ni37Fe25 and 133 °C for Cu38Ni37Al25 (Figure 8B). The decrease of the temperature of DTG_1_ as the copper content y increases was previously observed for a series of CuNiAl-LDHs with 0.16 ≤ y ≤ 0.81 [73]. This effect can be related to the larger ionic size of Cu^2+^ (0.73 Å) than Ni^2+^ (0.69 Å) [74]. The larger effective ion size of Cu^2+^, beyond increasing the cell parameter (Table 1), decreases the charge-to-size ratio of the divalent cation, contributing to a weakening of the interaction between the layers and the interlayer water molecules. It can be remarked in Table 2 that the decrease of the DTG_1_ temperature with the Cu content is less pronounced for Fe- than Al-bearing samples. A possible contribution of dehydroxylation occurring concurrently to dehydration in the former samples could explain this effect and will be considered by comparing the experimental and theoretical mass losses in the following.

The temperature of the DTG_2_ peak provides information on the ease of condensation of the layer hydroxyls, with decomposition of the LDH structure. Such a reaction usually occurs by a succession of intermediate steps. In the Ni-richest samples (Figure 8A) the DTG_2_ peak is quite sharp, with maxima at 294 and 321 °C for, respectively, Ni75Fe25 and Ni75Al25. For samples with an even content of Cu and Ni, the peaks are broader and more evenly split in components at 247 and 285 °C for Cu38Ni37Fe25 and at 250 and 305 °C for Cu38Ni37Al25 (Figure 8B). The growth of the 240–250 °C component accounts for an easier dehydroxylation in Cu-richer than in Ni-richer LDHs, which is consistent with the respective behaviour of Cu(OH)_2_ compared to Ni(OH)_2_ previously pointed out [59,60]. Moreover, the slightly lower DTG_2_ temperature in the Fe- than the Al-bearing samples of similar divalent cations distribution can be related to the larger cation radius of Fe^3+^ than Al^3+^ (0.64 vs. 0.54 Å) [74], with a corresponding lower charge-to-size ratio, which may weaken the bond strength of the hydroxyls in the layer and favour their condensation.

Better assignment of the experimental mass losses observed in the TG–DTG experiments can be obtained through a comparison of the values in the different steps with the theoretical ones corresponding to dehydration, dehydroxylation, and decarbonation, calculated from the LDH theoretical formula. The formulae of the LDHs, Cu_y(1−x)_Ni(_1−y)(1−x)_Al_x_(OH)_2_(CO_3_)_0.5x·_ zH_2_O, have been established assuming that CO_3_^2−^ is the only compensating anion, as previously observed. The amount of water molecules, z, has been calculated by subtracting calculated dehydroxylation and decarbonation losses from the experimental total weight loss. No theoretical formula of the LDH phase has been calculated for some samples: Cu68Ni07Al25 and Cu68Ni07Fe25, due to the contamination by tenorite, and for Cu75Fe25, due to the absence of LDH.

Experimental and calculated mass losses for each step of the activation are reported in Table 3 and compared in Figure 9.

In Figure 9, points on the diagonal correspond to the exclusive attribution of DTG_1_ mass loss to dehydration (Figure 9A), DTG_2_ to dehydroxylation (Figure 9B), and DTG_3_ and DTG_4_ to decarbonation (Figure 9C). Shifts from the diagonal can be described as following three basic patterns. Rhombohedral Al-bearing LDHs present a DTG_1_ mass loss corresponding to the theoretical dehydration amount (Figure 9A), whereas positive and negative deviations from the diagonal in, respectively, Figure 9B,C indicate that nearly one-third of carbonates are released in DTG_2_, together with dehydroxylation products. Indeed, a partial combination of dehydroxylation and decarbonation has been previously reported in carbonate-containing multicationic nickel- or copper-based LDHs [54,55,73,75,76].

The Fe-bearing rhombohedral LDHs share with Al-bearing samples a negative deviation of the decarbonation pattern (Figure 9C) but present a theoretical value of DTG_2_ (Figure 9B) and a positive deviation of DTG_1_ (Figure 9A). This suggests a more complex activation pattern, in which 40–60% decarbonation occurs in DTG_2_, superposed to dehydroxylation, but 5–12% dehydroxylation takes place in DTG_1_, together with dehydration.

The Cu-richest monoclinic Cu75Al25, highlighted by red arrows, presents a completely different pattern, with negative deviation in Figure 9B and positive deviations in Figure 9A,C. This indicates a more complex pattern of dehydroxylation, already started in DTG_1_, superposed to dehydration, and continuing in DTG_3_, together with decarbonation. Multidecomposition steps in the temperature range of 400–800 °C have been previously reported for CuAl-LDH, CuMnAl-LDH, and CuNiAl-LDH. Mass spectrometry analysis has shown that they correspond to the removal of carbonate species [54,55,58,73]. Their nature is controversial in the literature. They have been attributed to diverse species formed during thermal decomposition, such as oxycarbonates resulting from the bonding of CO_3_^2−^ to the hydroxide groups [54,73] or to precursor CuCO_3_ and Al_2_(CO_3_)_3_ species [58]. Alejandre et al. showed by IR spectroscopy that two types of carbonate species, giving rise to peaks at 1380 and 1500 cm^−1^, assigned respectively to monodentate and bidentate species, are present in the interlayer space of CO_3_-CuAl-LDH. They considered that the more stable bidentate species are decomposed above 430 °C [55]. Rives and Kannan underlined that the metastable phase formed at about 600 °C must be a pure copper-containing oxycarbonate, because it is formed from whatever bivalent or trivalent cation that is associated with copper [73].

The decomposition of the Cu-richest LDHs is a complex process characterized by a large superposition of dehydroxylation and decarbonation features. The TG and XRD analyses revealed progressive shrinkage of the LDH phases with complete decomposition at 300 °C and then concurrent dehydroxylation and decarbonation with crystallization of different oxide phases.

### 4.2. Formation of Oxide Phases by Crystallization of the Products of LDH Decomposition

In the temperature range from 300 to 800 °C, the effect of composition on the pattern of crystallization can be better appraised by comparing the amount of crystalline phases observed in the Cu-Ni series with Al or Fe trivalent cation, as reported in Figure 10. This allows focusing on the reactivity of the intermediate amorphous aluminium and iron oxide phases, a feature previously rarely discussed.

As a first general comment, it is worth emphasizing that the TG-DTG profiles (Figure 2 and Figure 6) have been recorded at a heating rate of 10 °C/min, faster than the heating rate of 5 °C/min of the XRD ramps. As a consequence, phenomena have more time to develop and are observed at a slightly lower temperature in the XRD ramps than in TG-DTG. For several samples, this explains that crystalline oxide phases have been present since 300 °C, despite complete dehydroxylation and decarbonation in most cases not in the TG-DTG profiles.

A better understanding of the different behaviours can be obtained considering separately the samples with a different structure of the parent phase, namely rhombohedral LDHs (Al- and Fe-bearing samples with Cu/(Cu + Ni) between 0 and 0.5), CuO-rich samples (Cu68Ni07Al25, Cu68Ni07Fe25 and Cu75Fe25), and monoclinic LDH (Cu75Al25).

Among the products of the decomposition of rhombohedral LDH, Ni75Al25 contains a nearly equal amount of crystalline NiO and amorphous phase in all the temperature ranges from 300 to 800 °C, revealing a remarkable thermal stability of the amorphous nickel–aluminium phase. The nickel-containing mixed oxides obtained by thermal treatment of NiAl LDH were extensively studied by Trifirò and coworkers [77,78,79,80]. They showed that, in addition to NiO, an amorphous Ni-doped alumina was formed, which evolved to NiAl_2_O_4_ spinel only by 15 h of calcination at a temperature higher than 750 °C. They attempted to quantify the amorphous alumina by dissolution in a NaOH solution and observed that the amount increased with the temperature of calcination up to 750 °C, where it reached 60% of the initial content. This suggested that the amorphous material became more soluble when enriched in Al by the exsolution of NiO upon thermal treatment [79]. The variation of the surface composition of the sample, before and after NaOH treatment, followed by X-ray fluorescence and XPS analysis, established that the remaining aluminium not dissolved by NaOH belonged to an amorphous “spinel-type” phase [78,80]. Furthermore, this latter was located at the interface between NiO and the Al-rich compound dissolved by NaOH. In summary, the thermal decomposition of NiAl LDH up to 750–800 °C leads to a mixture of NiO, probably doped with aluminium, together with a complex amorphous component, including a spinel-type phase and a nickel-doped alumina phase that can be removed by NaOH leaching. The spinel-type phase would play a major role in the thermal stability of NiO, as it would decorate or act as a support for the NiO particles [79]. Our results are also in agreement with a low thermal reactivity of the amorphous nickel–alumina phase, even at 800 °C. In the case of the Fe-bearing sample, Ni75Fe25, the formation of a small fraction of NiFe_2_O_4_ spinel at 700 °C is in good agreement with the literature data [81].

The retention of a similar amount of NiO and an amorphous nickel–trivalent phase throughout the temperature range is a feature unaltered by the replacement of 10% nickel by copper in Cu07Ni68Al25 or by the complete replacement of aluminium by iron in Ni75Fe25 (Figure 10). A broad peak at 35.4° 2θ can be observed in the XRD pattern of the two Al-bearing samples, Ni75Al25 and Cu07Ni68Al25, with a nearly constant intensity throughout the temperature range from 300 to 800 °C (Figure 4, Appendix A). This signal corresponds to a d = 2.93 Å. This interplanar spacing is too short to be attributed to the 311 reflection of an aluminate spinel [66] and seems related to at least a fraction of the nickel–aluminium amorphous phase. Under the assumption that a negligible amount of iron is incorporated in the NiO bunsenite phase, the composition of the amorphous phase in these Ni-rich samples can be evaluated at atomic ratios of Ni/trivalent ca. 1.

The samples with a larger amount of copper in the Al-bearing series and as much as 10% of copper in the Fe-bearing series present a significantly different pattern of crystallization. The amount of crystalline phases formed at 300 °C is smaller than in the Ni-rich samples and ranges from 12 to 25 and 37% in, respectively, Cu38Ni37Al25, Cu07Ni68Fe25, and Cu38Ni37Fe25 (Figure 10). Contrary to the amorphous phases formed in the Ni-rich samples that remain stable at a higher temperature, the amorphous phases in the Cu-richer samples are highly susceptible to crystallization when the temperature increases. The effect of copper incorporation is especially visible when the iron is the trivalent cation, with complete crystallization being reached at 800 °C in Cu07Ni68Fe25 and Cu38Ni37Fe25. The divalent/trivalent ratio in the amorphous phases at 300 °C can be evaluated at 2.3, 2, and 1.6, for, respectively, Cu38Ni37Al25, Cu07Ni68Fe25, and Cu38Ni37Fe25. These relatively high divalent cation contents probably contribute to the thermal reactivity of the amorphous phase. This reactivity is higher in the Fe-bearing sample needing lower divalent cation content. In these samples, a spinel phase appears at 600 °C and grows with temperature until a mass fraction corresponding to the total incorporation of iron in a spinel ferrite is reached at 800 °C. In the case of the Cu38Ni37Al25 sample, a nearly 37% amorphous phase is still present at 800 °C and no spinel is observed. The easier crystallization of the Fe-bearing amorphous phase is in agreement with the previously reported lower temperature of formation of ferrite than aluminate spinel [44].

The ratio between the NiO and CuO crystalline phases provides some hints about the differential incorporation of nickel and copper in the amorphous phase and in the ferrite. In the case of Cu07Ni68Fe25, no CuO tenorite is observed. In the samples with an even amount of nickel and copper, CuO tenorite is always in a lower amount than NiO bunsenite. These effects suggest that copper is preferentially retained in the amorphous phase directly formed by LDH decomposition and is also the major divalent cation in the spinel formed at a higher temperature.

The crystallization behaviour of Cu-Ni rhombohedral LDHs can be summarized by observing that the decomposition of the Ni-richest LDHs directly forms nanocrystalline NiO and leaves a thermally stable amorphous phase with similar amounts of divalent and trivalent cations. The introduction of copper hinders the direct formation of crystalline phases by the decomposition of the LDH structure. A larger amount of amorphous phase is formed, with a higher divalent cation content. This phase underwent easier crystallization when the temperature was raised. It has to be observed that the crystallization pattern of Fe-bearing samples is more affected by the introduction of lower amounts of copper than the Al-bearing samples.

The crystallization pattern of the Cu-richest samples is affected by the presence of CuO tenorite. The effect is especially remarkable in Cu75Fe25 and Cu68Ni07Fe25, where tenorite is the only or major phase formed in the precipitation-ageing process. Tenorite represents nearly half of the mass of samples at 300 °C and is accompanied by an amorphous copper–iron phase with a divalent/trivalent cation ratio of ca. 1. The amount of the amorphous phase is remarkably stable in the temperature range from 300 to 500 °C. At a higher temperature, the spinel phase appears and increases until only ca. 15% trivalent cation-richer amorphous phase is left at 800 °C.

Some tenorite is also formed in the precipitation-ageing process of Cu68Ni07Al25, but it represents only a minor phase. The amount of tenorite in the sample is 25% at 300 °C and increases beyond 600 °C. Nearly 30% of the amorphous phase is left at 800 °C, and spinel appears only at this temperature, confirming that the aluminate spinels are less easily formed than ferrites.

Among the Cu-richest samples, Cu75Al25 follows a special crystallization pattern, as the precipitation product does not contain any tenorite, and the only crystalline phase is monoclinic LDH. This confirms that the monoclinic LDH phase is more effective than the rhombohedral phase in the inclusion of copper cations in the lattice. On the decomposition of the LDH, a small amount of nearly 10% tenorite is the only crystalline phase formed at 300 °C. The 90% amorphous copper–aluminium phase is remarkably stable when the temperature increases and the crystallization of more tenorite begins just at 600 °C. Copper aluminate spinel appears at 700 °C, and a virtually complete crystallization to stoichiometric amounts of tenorite and spinel is reached at 800 °C.

Globally, the thermal ramp data confirm the high dependence on the composition of the thermal reactivity of the amorphous phase formed at the decomposition of the LDH. As often observed, only divalent cation-bearing crystalline phases are formed at a low temperature, already upon precipitation in the case of copper-rich samples and at the decomposition of LDH in the case of Ni-rich samples. The crystallization of divalent cation oxides at a low temperature is hindered in samples with mixed divalent cations. In these samples, NiO bunsenite is preferentially formed at the expense of CuO tenorite. However, the presence of copper increases the reactivity of the amorphous phase formed at temperatures higher than 600 °C. On the contrary, the fraction of trivalent cations in the amorphous phase increases its thermal stability, especially when aluminium is the trivalent cation.

## 5. Conclusions

The results obtained in this work provide a better understanding of the role played by the different types of divalent and trivalent cations on the decomposition processes of (Cu,Ni)Al- and (Cu,Ni)Fe-LDH precursors and the crystallization of the mixed oxides formed upon thermal activation.

The Ni-richer LDH precursors exhibit rhombohedral structures with either Al or Fe trivalent cations while the Cu-richer LDH precursors exhibit a monoclinic structure with Al and the concurrent formation of rhombohedral LDH and tenorite with Fe as trivalent cation. Tenorite is only present when Cu and Fe are coprecipitated showing that these cations cannot be combined into an LDH structure.

The influence of the cationic composition on the decomposition processes of the LDHs is evidenced by TG–DTG and in situ XRD analyses. In all cases, the dehydration, dehydroxylation, and decarbonation features classically observed are differently connected. For the rhombohedral Al-bearing LDHs, dehydration alone first occurs. Further, one-third of carbonates is released together with the dehydroxylation products showing partial recovery of dehydroxylation and decarbonation. In addition, this recovery is more pronounced in the Fe-bearing rhombohedral LDHs, with dehydration containing 5–12% of dehydroxylation species and 40–60% of decarbonation occurring with dehydroxylation. A much higher recovery of the three different features is even observed for the monoclinic CuAl LDH. Accordingly, the crystallization of the oxides in the temperature range from 300 to 800 °C emphasizes the major role played both by the nature of the divalent and trivalent cations. Regarding the influence of the divalent cation, the mixed Ni-Cu amorphous phase is more reactive than the Cu-free amorphous phase, which is mainly related to the twofold higher divalenton trivalent cation ratio in the former than the latter. Consistently, the Ni-richer rhombohedral LDHs with a Ni–trivalent atomic ratio of ca. 1 lead to the formation, in all the temperature ranges, of an even amount of NiO and a nickel–aluminium amorphous phase. The increase in the divalent ontrivalent cation ratio to ca. 2 upon the introduction of copper in the rhombohedral LDHs inhibits the direct crystallization of the oxide phases at a low temperature (300–500 °C) but, on the contrary, greatly improves the crystallization above 600 °C. Regarding the influence of the trivalent cation, the crystallization is improved in the Fe-bearing samples and the formation of spinel phases occurs at a lower temperature in comparison to the Al-bearing samples. Rather different crystallization behaviour is observed for the Cu-richest monoclinic CuAl LDH structure. A remarkably high and stable amount of the amorphous copper–aluminium phase is present until 500 °C. Then, the tenorite amount increases and the copper–aluminate spinel phase crystallizes at 700 °C. Therefore, these results reveal that the reactivity is higher for the amorphous iron than aluminium phases and is proportional to the amount of embedded divalent cations. They provide useful information to optimize the activation of the ternary (Cu,Ni)Al- and (Cu,Ni)Fe-LDH for several types of applications, particularly as catalytic materials.

## Figures and Tables

**Figure 1 materials-17-00083-f001:**
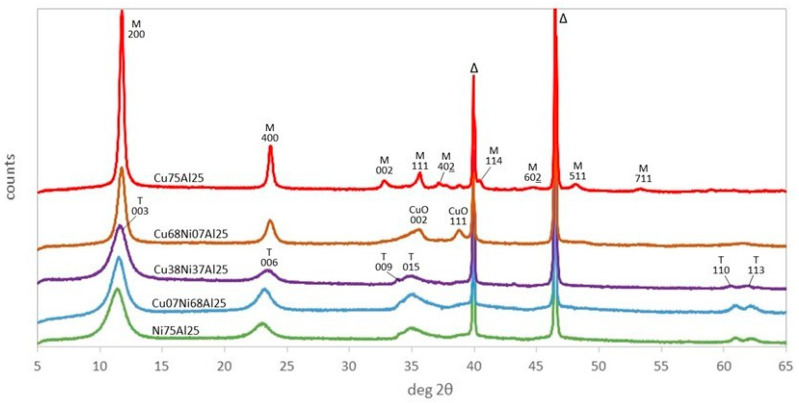
XRD patterns for as-synthesized (Cu,Ni)Al samples. Indexed peaks: M, monoclinic and T, trigonal LDHs, CuO tenorite, Δ: PtRh powder support.

**Figure 2 materials-17-00083-f002:**
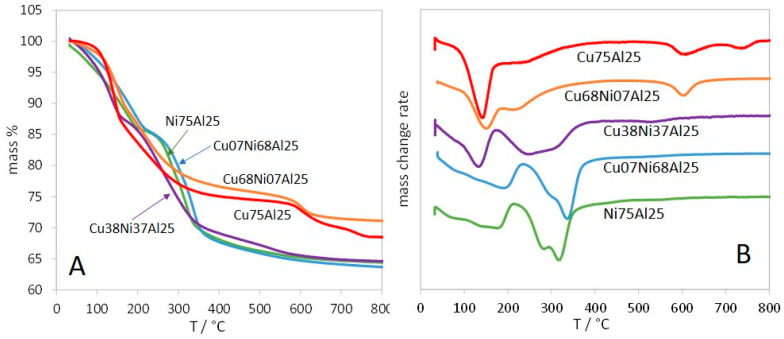
TG (**A**) and DTG (**B**) curves of the (Cu,Ni)Al samples.

**Figure 3 materials-17-00083-f003:**
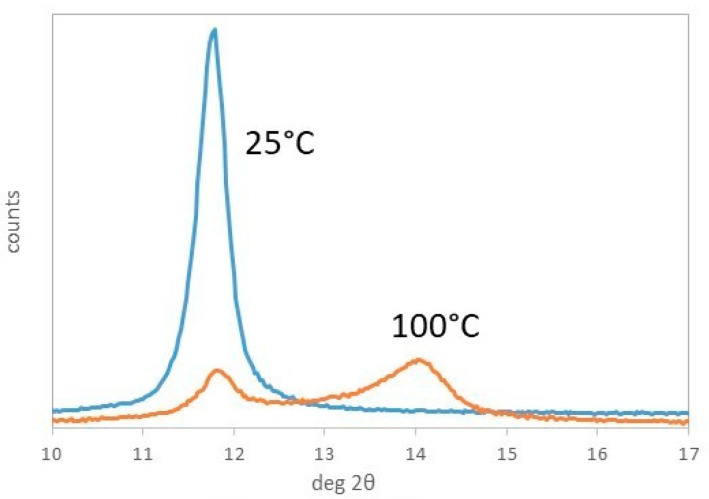
XRD patterns of sample Cu75Al25 at room temperature and 100 °C.

**Figure 4 materials-17-00083-f004:**
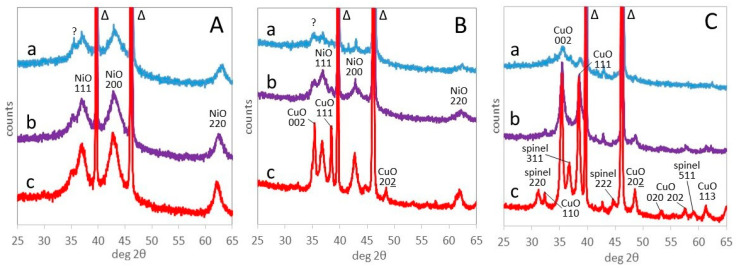
XRD patterns of Ni75Al25 (**A**), Cu38Ni37Al25 (**B**), and Cu75Al25 (**C**) at 400 (a), 600 (b) and 800 °C (c). ?: unknown (see text). Indexed peaks: NiO bunsenite, CuO tenorite, spinel, Δ: PtRh powder support.

**Figure 5 materials-17-00083-f005:**
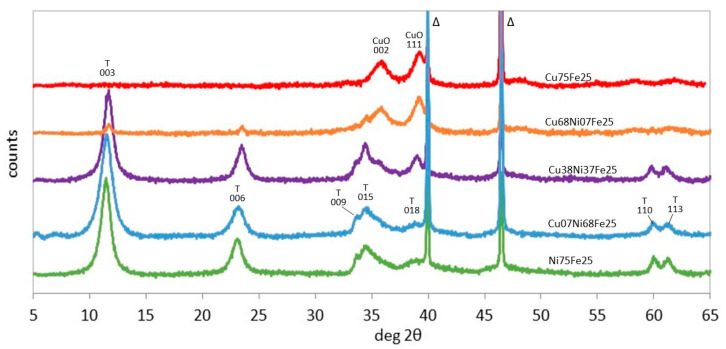
XRD patterns of as-synthesized (Cu,Ni)Fe samples. Indexed peaks: T, trigonal LDH, CuO tenorite, Δ: PtRh powder support.

**Figure 6 materials-17-00083-f006:**
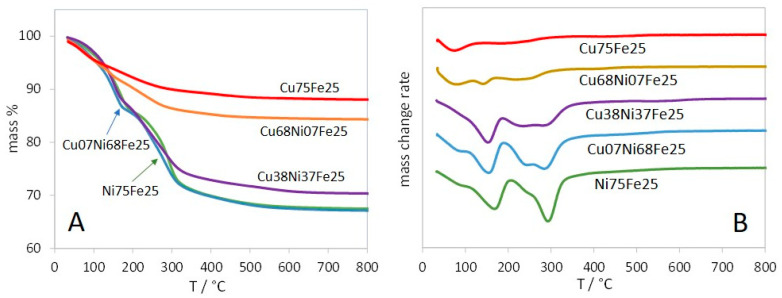
TG (**A**) and DTG (**B**) curves of (Cu,Ni)Fe samples. The DTG curves are shifted for the sake of clarity.

**Figure 7 materials-17-00083-f007:**
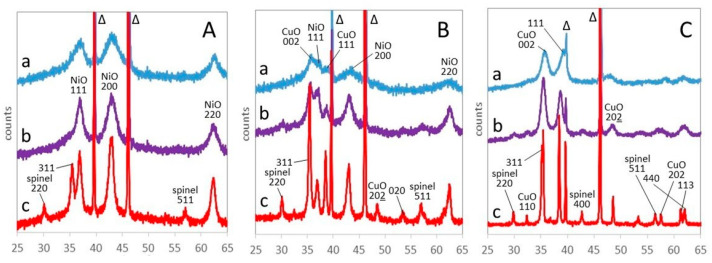
XRD patterns of Ni75Fe25 (**A**), Cu38Ni37Fe25 (**B**), and Cu75Fe25 (**C**) at 400 (a), 600 (b), and 800 (c) °C. Indexed peaks: NiO bunsenite, CuO tenorite, spinel, Δ: PtRh powder support.

**Figure 8 materials-17-00083-f008:**
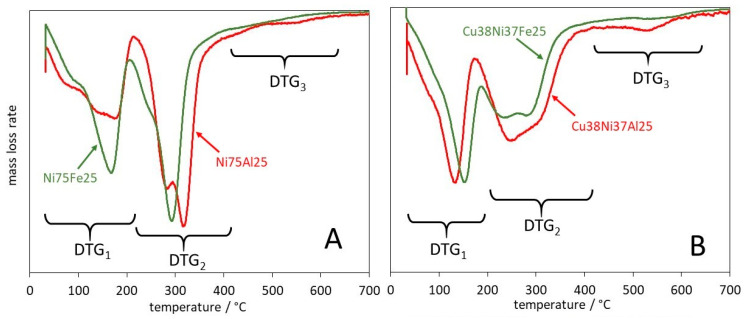
Comparison of the DTG curves of Al-bearing (red traces) and Fe-bearing (green traces) samples with Cu/(Cu + Ni) = y = 0 (**A**) or 0.5 (**B**). DTGn labels as in Table 2.

**Figure 9 materials-17-00083-f009:**
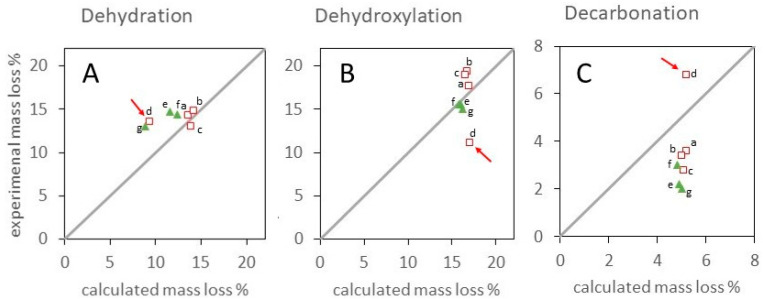
Comparison of experimental and theoretical mass losses in DTG_1_ (**A**), DTG_2_ (**B**), and DTG_3_ + DTG_4_ (**C**) steps from Table 3 for Al-bearing (void squares) and Fe-bearing (green triangles) samples. Samples Ni75Al25 (a), Cu08Ni67Al25 (b), Cu38Ni37Al25 (c), Cu75Al25 (d), Ni75Fe25 (e), Cu07Ni68Fe25 (f), and Cu38Ni37Fe25 (g). A red arrow highlights monoclinic Cu75Al25.

**Figure 10 materials-17-00083-f010:**
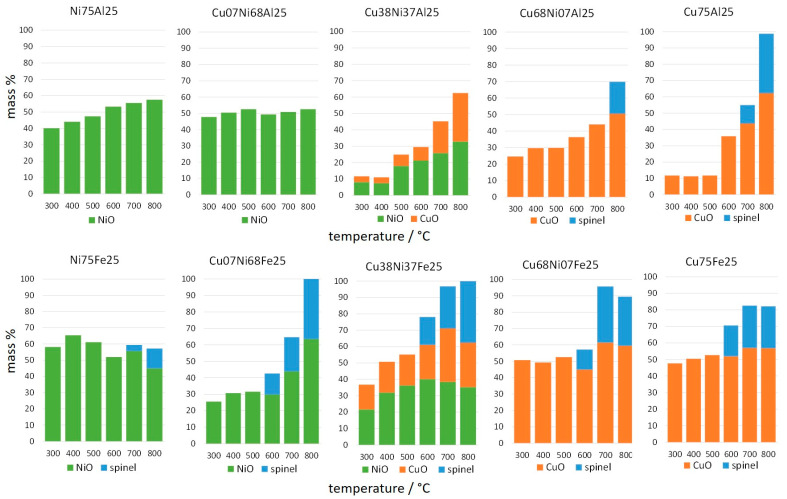
Mass % of phases in samples of the series (Cu,Ni)Al (**top**) and (Cu,Ni)Fe (**bottom**) as a function of temperature in XRD thermal ramp: NiO bunsenite (green), CuO tenorite (red), and spinel (blue).

**Table 1 materials-17-00083-t001:** Crystallographic data of the LDH phases.

Sample	d003 (Å)			a (Å)
	25 °C	100 °C	200 °C	25 °C
Ni75Al25	7.61	7.39	6.58	3.040
Cu07Ni68Al25	7.63	7.39	6.53	3.040
Cu38Ni37Al25	7.52	6.62	6.42	3.059
Cu68Ni07Al25 *	7.52	6.52	5.64	-
Cu75Al25 *	7.52 -	7.49 6.31	- 5.61	- -
Ni75Fe25	7.73	7.14	6.41	3.082
Cu07Ni68Fe25	7.73	6.73	6.37	3.086
Cu38Ni37Fe25	7.59	6.46	5.5	3.093
Cu68Ni07Fe25	7.56	6.39	-	-
Cu75Fe25	-	-	-	-

* monoclinic d200.

**Table 2 materials-17-00083-t002:** Temperature range, weight losses, and maximum temperature of DTG peaks for the decomposition steps of the (Cu,Ni)Al and (Cu,Ni)Fe series of samples obtained by TGA.

**Sample**	**Phases**	**1st Step**			**2nd Step**			**3rd Step**			**4th Step**			**Total** **Loss** **(%)**
		**(°C)**	**∆m_1_** **(wt%)**	**DTG_1_** **(°C)**	**T** **(°C)**	**∆m_2_** **(wt%)**	**DTG_2_** **(°C)**	**T** **(°C)**	**∆m_3_** **(wt%)**	**DTG_3_** **(°C)**	**T** **(°C)**	**∆m_4_** **(wt%)**	**DTG_4_** **(°C)**	
Ni75Al25	LDH	25–213	14.3	85_w_ 140_w_ 177_s_	213–400	17.7	283_m_ 321_s_	400–700	3.6	545				35.6
Cu07Ni68Al25	LDH	25–235	14.8	196	235–444	18.4	310_m_ 337_s_	444–800	2.8	510				36
Cu38Ni37Al25	LDH	25–175	13	80_w_ 133_s_	175–450	19	251_s_ 305_m_	450–700	3.4	527				35.4
Cu68Ni07Al25	LDH, CuO_w_	25–165	10	152	165–450	14	213	450–750	5	601				29
Cu75Al25	LDH	25–167	13.6	142	167–400	11.1	240	400–680	5.2	604	680–800	1.6	737	31.5
Ni75Fe25	LDH	25–210	14.7	96_w_ 168_s_	210–405	15.6	250_w_ 294_s_	405–650	2.2	-				32.5
Cu07Ni68Fe25	LDH	25–195	14.4	92_w_ 154_s_	195–390	15.6	247_w_ 285_s_	390–600	3	-				33
Cu38Ni37Fe25	LDH	25–190	13	91_w_ 153_s_	190–470	15	243_m_ 283_m_	470–800	2	-				30
Cu68Ni07Fe25	CuO, LDH_w_	25–115	5.4	82	115–365	9.1	143_s_ 230_s_	365–600	1.5	405				16
Cu75Fe25	CuO	25–134	6	77	134–350	4.7	190	350–800	1.3	427				12

W: weak; M: medium; S: strong.

**Table 3 materials-17-00083-t003:** Comparison of experimental mass losses in the different decomposition steps with calculated values based on sequential dehydration, dehydroxylation, and decarbonation.

Sample	LDH Formula	∆m Experimental (%)					∆m Theoretical (%)		
		Step 1	Step 2	Step 3	Step 4	Total	Dehydration	Dehydroxylation	Decarbonation
Ni75Al25	Ni_0.75_Al_0.25_(OH)_2_(CO_3_)_0.125_·0.81 H_2_O	14.3	17.7	3.6	-	35.6	13.6	16.9	5.2
Cu08Ni67Al25	Cu_0.08_Ni_0.67_Al_0.25_(OH)_2_(CO_3_)_0.125_·0.85 H_2_O	14.8	19.4	2.8	-	36	14.2	16.7	5.1
Cu38Ni37Al25	Cu_0.38_Ni_0.37_Al_0.25_(OH)_2_ (CO_3_)_0.125_·0.84 H_2_O	13	19	3.4	-	35.4	13.9	16.5	5
Cu75Al25	Cu_0.75_ Al_0.25_(OH)_2_ (CO_3_)_0.125_·0.54 H_2_O	13.6	11.1	5.2	1.6	31.5	9.3	17	5.2
Ni75Fe25	Ni_0.75_Fe_0.25_(OH)_2_ (CO_3_)_0.125_·0.73 H_2_O	14.7	15.6	2.2	-	32.5	11.6	16	4.9
Cu07Ni68Fe25	Cu_0.07_Ni_0.68_ Fe_0.25_(OH)_2_ (CO_3_)_0.125_·0.78 H_2_O	14.4	15.6	3	-	33	12.4	15.8	4.8
Cu38Ni37Fe25	Cu_0.38_Ni_0.37_ Fe_0.25_(OH)_2_ (CO_3_)_0.125_·0.55 H_2_O	13	15	2	-	30	8.8	16.2	5

Cu68Ni07Al25, Cu68Ni07Fe25, and Cu75Fe25 have not been modelled as LDH, due to the presence of other phases.

## Data Availability

Data are contained within the article and Appendix A.

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
