# Peer review of "Composition Effect on the Formation of Oxide Phases by Thermal Decomposition of CuNiM(III) Layered Double Hydroxides with M(III) = Al, Fe"

_materials, 2023, doi:10.3390/ma17010083_

Round 1
Reviewer 1 Report
Comments and Suggestions for Authors
The authors replied to my previous comments:
"However before resubmission the authors should correct the manuscript by including the results related to the calculation of the interlayer distance. Table 1 presented in the manuscript contains only the data refering to "a" parameter which are related to the distance between cations in the brucite type layer and is determined only at 25oC. Meanwhile, d003 value is presented at 25, 100 and 200o C, but the value of c parameter related to the interlayer distance is not presented but it is commented throughout the text."
Author's answer - This part has been modified
Looking at their current submission I see that Table 1 is identical to the one in their previous submission
Could you please check again?
Comments on the Quality of English LanguageThe manuscript contains only minor errors in English and several mistyped letters
line 100 - established instead of establish
line 119 - the kinetics for the growth of the different crystalline phases (add the red text)
line 127 Ni(NO3)2.6H2O
line 128 Cu(NO3)2.3H2O
line 128 Al(NO3)3.9H2O
line 129 Fe(NO3)3.9H2O
Reviewer 2 Report
Comments and Suggestions for Authors
The aim of the work is not clear from the introduction.
line 127: Chemical formula of Nickel (II) nitrate hexahydrate should be corrected.
line 128: Chemical formula of nitrate trihydrate and aluminum nitrate nonahydrate should be corrected.
line 129: Chemical formula of nitrate nonahydrate should be corrected.
line 156: The sentence "Samples were named by their percent atomic fraction of cations, for instance, Cu38Ni37Fe25." was previously used in line 143. I suggest deleting.
line 291: ... Scherrer equation is only valid for cubic crystallite. However, the samples of this work have other symmetries. Please correct.
Conclusions should be shorter and focused on meeting the objectives of the work.
Reviewer 3 Report
Comments and Suggestions for Authors
In their work, Awan and colleagues describe the effect of composition on the formation of oxide phases in layered double hydroxides with the inclusion of Al and Fe. Their work primarily consists of thermogravimetry and X-ray diffraction measurements. Their work presents a great deal of results for a wide range of materials, but I have some specific concerns that I would like to see addressed.
1.) In figure 1, what is the reason for additional peaks in Cu75Al25 located at ~32.5 degrees, between 35 and 40, and at about 47? Can the authors index the observed XRD peaks? In fact, I see no indexing performed at all, throughout this work. Further, can the authors indicate a reference (or a JCPDS card, etc) that can be used to identify their material/show it matches the stated structure?
2.) Again, in figure 4, it would be useful to show what peaks correspond to which material and which crystal plans within the figure itself.
3.) Line 314 states "A quantitative distribution of Ni2+ and Al3+ cations in the different phases of Ni75Al25 314 is difficult to ascertain." Have the authors considered performing measurements such as XPS, which may help here?
4.) Figure 9 is a bit confusing. Cu75Al25 is highlighted by a red arrow, but it may be more useful to label more of the individual points, so readers know what points correspond to which samples.
5.) Around line 520, authors state "As a first general comment, it is worth emphasizing that the TG-DTG profiles (Figures 2 and 6) have been recorded at a heating rate of 10 °C/min, faster than the heating 520 rate of 5°C/min of the XRD ramps." - why were different rates chosen?
Now a few general comments.
1.) Figure quality is too low/pixelated, and it would be better to use higher resolution images.
2.) A lot of the discussion seems rather speculative. Can the authors comment?
I do think the work has a wide range of very systematic measurements across this class of materials. However, I also think the science isn't fully clear - the XRD analysis needs to be better explained/more clearly discussed (i.e. indicate where peaks arise from in the images), and it's possible more information could be gleaned from further analysis using measurments like XPS/EDX/etc.
Round 2
Reviewer 3 Report
Comments and Suggestions for Authors
The authors have sufficiently addressed my concerns.